# Post-Disaster Building Damage Detection from Earth Observation Imagery Using Unsupervised and Transferable Anomaly Detecting Generative Adversarial Networks

**Sofia Tilon** \*, **Francesco Nex**, **Norman Kerle** **and George Vosselman**

Faculty of Geo-Information Science and Earth Observation (ITC), University of Twente, 7514 AE Enschede, The Netherlands; f.nex@utwente.nl (F.N.); n.kerle@utwente.nl (N.K.); george.vosselman@utwente.nl (G.V.)

\* Correspondence: s.m.tilon@utwente.nl

**Abstract:** We present an unsupervised deep learning approach for post-disaster building damage detection that can transfer to different typologies of damage or geographical locations. Previous advances in this direction were limited by insufficient qualitative training data. We propose to use a state-of-the-art Anomaly Detecting Generative Adversarial Network (ADGAN) because it only requires pre-event imagery of buildings in their undamaged state. This approach aids the post-disaster response phase because the model can be developed in the pre-event phase and rapidly deployed in the post-event phase. We used the xBD dataset, containing pre- and post- event satellite imagery of several disaster-types, and a custom made Unmanned Aerial Vehicle (UAV) dataset, containing post-earthquake imagery. Results showed that models trained on UAV-imagery were capable of detecting earthquake-induced damage. The best performing model for European locations obtained a recall, precision and F1-score of 0.59, 0.97 and 0.74, respectively. Models trained on satellite imagery were capable of detecting damage on the condition that the training dataset was void of vegetation and shadows. In this manner, the best performing model for (wild)fire events yielded a recall, precision and F1-score of 0.78, 0.99 and 0.87, respectively. Compared to other supervised and/or multi-epoch approaches, our results are encouraging. Moreover, in addition to image classifications, we show how contextual information can be used to create detailed damage maps without the need of a dedicated multi-task deep learning framework. Finally, we formulate practical guidelines to apply this single-epoch and unsupervised method to real-world applications.

**Keywords:** deep learning; Generative Adversarial Networks; post-disaster; building damage assessment; anomaly detection; Unmanned Aerial Vehicles (UAV); satellite; xBD

## 1. Introduction

Damage detection is a critical element in the post-disaster response and recovery phase [1]. Therefore, it has been a topic of interest for decades. Recently, the popularity of deep learning has sparked a renewed interest in this topic [2–4].

Remote sensing imagery is a critical tool to analyze the impacts of a disaster in both the pre- and post-event epoch [4]. Such imagery can be obtained from different platforms: satellites, Unmanned Aerial Vehicles (UAV's) and manned aircrafts [5,6]. Each contains characteristics that need to be considered before deciding on which to use for disaster analysis. Manned airplanes or UAV's can be flexibly deployed and fly at relatively low heights compared to satellites and, therefore, have relatively small ground sampling distances (GSD) [7]. UAV's can fly lower than manned airplanes and in addition, depending on the type of drone, they can hover and maneuver in between obstacles.

Both platforms can be equipped with a camera in oblique mounts, meaning that vital information can be derived from not only the top but also the sides of objects [8]. However, data acquisitions using these platforms have to be carried out and instigated by humans, which makes them time and resource costly. The spatial coverage of these platforms is also typically restricted to small areas of interests (AIO) and biased towards post-event scenarios when new information is required. Therefore, pre-event data from UAV or aerial platforms are less likely to exist. Satellites on the other hand, depending on the type of satellite, have a high coverage and return rate, especially of build-up areas. Therefore, pre-event data from satellites are more likely to exist. Moreover, satellite systems that provide information to Emergency Mapping Services are able to (re)visit the disaster location only hours after an event, enabling fast damage mapping [9]. A disadvantage of satellite imagery is that it has larger GSD footprints. Moreover, excluding the ones that are freely available, obtaining satellite imagery is generally more costly than UAV imagery.

Damage mapping using Earth observation imagery and automatic image analysis is still a challenge for various reasons despite decades of dedicated research. Traditional image analysis remains sensitive to imaging conditions. Shadows, varying lighting conditions, temporal variety of objects, camera angles or distortions of 3D objects that have been reduced to a 2D plane have made it difficult to delineate damage. Moreover, the translation of found damage features to meaningful damage insights have prevented many methods from being implemented in real-world scenarios. Deep learning has made a major contribution towards solving these challenges by allowing the learning of damage features instead of handcrafting them. Several studies have been carried out on post-disaster building damage detection using remote sensing imagery and deep learning [6,10–13]. Adding 3D information, prior cadastral information or multi-scale imagery has contributed towards some of these challenges [11,14–16]. Despite these efforts, persistent problems related to vegetation, shadows or damage interpretation remain. More importantly, a lesser addressed aspect of deep learning-based post-disaster damage detection remains—the transferability of models to other locations or disasters. Models that can generalize and transfer well to other tasks constitute the overarching objective for deep learning applications. Specifically, in the post-disaster management domain, such a model would remove the need to obtain specific training data to address detection tasks for a particular location or disaster. By removing this time-costly part of post-disaster damage detection, resources are saved and fast post-disaster response and recovery is enabled. However, a persisting issue keeping this goal out of reach is the availability of sufficient qualitative training data [13].

Because disasters affect a variety of locations and objects, damage induced by disasters similarly shows a large variety in visual appearances [13]. Obtaining a number of images that sufficiently cover the range of visual appearances is difficult and impractical. In fact, it is impossible to sample the never before seen damage, making supervised deep learning models inherently ad hoc [17]. Moreover, it is challenging to obtain qualitative annotations. Ideally, images are labelled by domain experts. However, the annotation process is time-costly, which critical post-disaster scenarios cannot tolerate. Finally, the process is subjective. Especially in a multi-classification task, two experts are unlikely to annotate all samples with the same label [18]. Questionable quality of the input data makes it difficult to trust the resulting output. The problem of insufficient qualitative training data drives most studies to make use of data from other disaster events with damage similar to the one of interest, to apply transfer learning or to apply unsupervised learning [19].

Most unsupervised methods for damage detection are not adequate for post-disaster applications where time and data are scarce. Principal Component Analysis (PCA) or multi-temporal deep learning frameworks are used for unsupervised change detection [20,21]. Besides the disadvantage of PCA that it is slow and yields high computational overhead, a major disadvantage of change detection approaches in general is that pre-event imagery is required, which is not always available in post-disaster scenarios. Methods such as One-Class Support Vector Machines (OCSVM) make use of a single epoch; however, these methods cannot be considered unsupervised because the normal class, in this case the undamaged class, still needs to be annotated in order to distinguish anomalies such as damage [22]. Moreover,

earlier work has shown that OCSVM underperforms in the building damage detection task compared to supervised methods [23].

Anomaly detecting Generative Adversarial Networks (ADGANs), a recently proposed unsupervised deep learning principle used for anomaly detection, have the potential to overcome the aforementioned limitations and, therefore, to improve model transferability. ADGANs have been applied to detect anomalies in images that are less varied in appearance to address problems in constrained settings. For example, reference [17], reference [24] and reference [25] have applied ADGANs to detect prohibited items in x-rays of luggage. Reference [26] and reference [27] have applied ADGANs to detect masses in ultrasounds or disease markers in retina images. Until recently, ADGANs had not been applied to detect anomalies in images that are visually complex, such as remote sensing images, to address a problem that exists in a variety of settings, such as damage detection from remote sensing images.

The fundamental principle of an ADGAN is to view the damaged state as anomalous, and the undamaged state as normal. It only requires images that depict the normal, undamaged state. This principle poses several advantages. First, obtaining images from the undamaged state is less challenging, assuming that this state is the default. Second, data annotations are not required, thus eliminating the need of qualitative annotated training data. Finally, the never before seen damage is inherently considered since it deviates from the norm. This makes ADGAN an all-encompassing approach. The aforementioned advantages have made ADGANs appealing for a variety of applications, and especially appealing for post-disaster damage detection. The main advantage for post-disaster applications is that a model can be trained pre-disaster using only pre-event imagery. It can be instantly applied after the occurrence of a disaster using post-event imagery and thus aid post-disaster response and recovery. ADGANs output binary damage classifications and, therefore, a disadvantage is that they are unable to distinguish between damage severity levels. However, we argue that the practical advantages listed above outweigh this disadvantage, especially considering how the method provides rapid information to first responders in post-disaster scenes.

In earlier work, we showed how an ADGAN could be used under certain pre-processing constraints to detect post-earthquake building damage from imagery obtained from a manned aircraft [23]. Considering these results, and in addition the characteristics of the different remote sensing platforms explained above, we extend the preliminary work by investigating the applicability of ADGAN to detect damage from different remote sensing platforms. By training the ADGAN on a variety of pre-disaster scenes, we expect it to transfer well to different geographical locations or typologies of disasters. Special attention is given to satellite imagery because of its advantages explained above. We aim to provide practical recommendation on how to use this method in operational scenarios.

The contribution of this paper is threefold:

- First, we show how an ADGAN can be applied in a completely unsupervised manner to detect post-disaster building damage from different remote sensing platforms using only pre-event imagery.
- Second, we show how sensitive this method is against different types of pre-processing or data selections to guide practical guidelines for operational conditions.
- Lastly, we show whether this method can generalize over different typologies of damage or locations to explain the usability of the proposed method to real world scenarios.

The goal of this research is the fast detection of damage enabling fast dissemination of information to end-users in a post-disaster scenario. Therefore, it is beyond the scope of this study to examine the link between the proposed method and pre-event building vulnerability estimations or fragility curves. Our main aim is to investigate the applicability of ADGANs for unsupervised damage detection. Based on our results, we present a conclusion regarding the applicability and transferability of this method from an end-user's perspective.

Related work can be found in Section 2; the experiments are detailed in Section 3; results are described in Section 4; the discussion and conclusion can be found in Sections 5 and 6, respectively.

## 2. Related Work

### 2.1. Deep Learning for Post-disaster Damage Detection

Deep learning using optical remote sensing imagery has been a widely researched topic to address various aspects in the post-disaster research domain. Reference [2] used a SqueezeNet based Convolutional Neural Net (CNN) to make a distinction between collapsed and non-collapsed buildings after an earthquake event. Reference [28] addressed the combined use of satellite and airborne imagery at different resolutions to improve building damage detection. Reference [12] proposed a method to detect different proxies of damage, such as roof damage, debris, flooded areas, by using transfer learning and airborne imagery. Similarly, Reference [3] aimed to detect blue tarp covered buildings, a proxy for building damage, by utilizing aerial imagery and building footprints. Various researchers focused on utilizing pre- and post-event imagery to its best advantage. Reference [29] showed how fusion of multi-temporal features improved damage localization and classification. Similarly, reference [30] aimed to detect different building damage degrees by evaluating the use of popular CNNs and multi-temporal satellite imagery. Reference [11] proposed an efficient method to update building databases by using pre-disaster satellite imagery and building footprints to train a CNN, which was fine-tuned using post-disaster imagery. Reference [31] proposed a U-Net-based segmentation model to segment roads and buildings from pre- and post-disaster satellite imagery, specifically to update road networks. Progress has also been made towards real-time damage detection. Reference [32] made use of a lightweight CNN that was placed on board an UAV to detect forest fires in semi-real time. Reference [7] developed a similar near-real time low-cost UAV-based system which was able to stream building damage to end-users on the ground. Their approach was one of the first to validate such a system in large-scale projects. Finally, reference [14] showed how adding 3D information to UAV imagery aided the detection of minor damage on building facades from oblique UAV imagery.

Most deep learning methods towards post-disaster damage mapping, including the ones mentioned above, are supervised. However, a persistent issue in supervised learning is the lack of labelled training data [4]. The issue of unbalanced datasets or the lack of qualitative datasets is mentioned by most [2,12,28–30]. As mentioned earlier, researchers bypass this issue by using training datasets from other projects that resembles the data that are needed for the task-at-hand, or by applying transfer learning to boost performance. Despite these solutions, the main weakness of these solutions is that these models generally do not transfer well to other datasets. Reference [13] compared the transferability of different CNNs that were trained on UAV and satellite data from different geographic locations, and concluded that the data used for training a model strongly influences the model its ability to transfer to other datasets. Therefore, especially in data scarce regions, the application of damage detection methodologies in operative scenarios remains limited.

### 2.2. Generative Adversarial Networks

Generative Adversarial Networks (GANs) were developed by reference [33] and gained popularity due to their applicability in a variety of fields. Applications include augmented reality, data generation and data augmentation [34–36]. A comprehensive review of research towards GANs from recent years can be found in reference [37].

A GAN consists of two Convolutional Neural Nets (CNNs): the Generator and the Discriminator. The Generator receives as input an image dataset with data distribution $p_{data}$. The Generator aims to produce a new image ($\hat{x}$) that fits within the distribution $p_{data}$. Therefore, the Generator aims to learn a distribution of $p_g$ that approaches $p_{data}$. The Discriminator receives as input an image ($x$) from the original dataset as well as the image ($\hat{x}$) generated by the Generator. The goal of the Discriminator is to distinguish the generated images from the original input data. If the Discriminator wins, the Generator

loses and vice versa [33]. The Generator (G) and Discriminator (D) are locked in the two-player zero-sum principle. The discriminator aims to minimize the function $D(G(x))$ and the Generator tries to maximize it according to the function $\log(1 - D(G(x)))$.

### 2.3. Anomaly detecting Generative Adversarial Networks.

GANs are also applied to detect defects or damage in the medical or manufacturing domain. Similar to post-disaster damage detection, a common limitation for these kind of applications is data imbalance. Therefore, GANs are used to synthesize more data of the underrepresented class. Reference [38] synthesized medical imagery to boost liver lesion detection and reference [39] synthesized road defects samples, which led to a F1-score increase of up to 5 percent. The main limitation of synthesizing data is that examples are required. Moreover, it is unclear to what extent the generated samples are restricted to the data distribution of the input data, inhibiting diversity of the generated images [40,41]. ADGANs provide a better solution, since no examples are needed.

ADGANs are only trained using normal, non-damaged input data. The resulting trained model is proficient in reproducing images that do not show damage, and less proficient in reproducing images that depict damage. Therefore, the distance between the input image and the generated image is large when inference is done using an image that contains damage, which subsequently can be used to produce anomaly scores [24].

The first examples of ADGANs are Efficient GAN-Based Anomaly Detection (EGBAD), which was developed using curated datasets such as MNIST, and AnoGAN, which was geared towards anomaly detection in medical imagery [27,42]. Reference [26] applied an EGBAD-based method to detect malign masses in mammograms. The main limitation in AnoGAN was its low inference speed. This was resolved in f-AnoGAN [43]. The latter was outperformed by GANomaly, which successfully detected prohibited items in x-rays of luggage [17], although it was shown to be less capable of reconstructing visually complex images [23,44]. Using a U-Net as the Generator, the reconstruction of complex imagery was resolved by its successor Skip-GANomaly [24]. Both f-AnoGAN and Skip-GANomaly serve as the basis for ongoing developments [25,44–46].

Considering that Skip-GANomaly outperformed f-AnoGAN, and in addition, how it is proficient in generating visually complex imagery, this architecture was used in this research.

## 3. Materials and Methods

### 3.1. ADGAN

The architecture of Skip-GANomaly is shown in Figure 1. The Generator and the Discriminator consist of a U-net and an encoder architecture, respectively [47]. In earlier work, we showed how substituting the Generator for an encoder–decoder architecture without skip-connections—e.g., GANomaly [17]—does not always result in well-reconstructed fake images from Earth observation imagery [23]. The encoder–decoder architecture of Skip-GANomaly, in combination with the skip-connections, makes it efficient in recreating even complex remote sensing imagery.

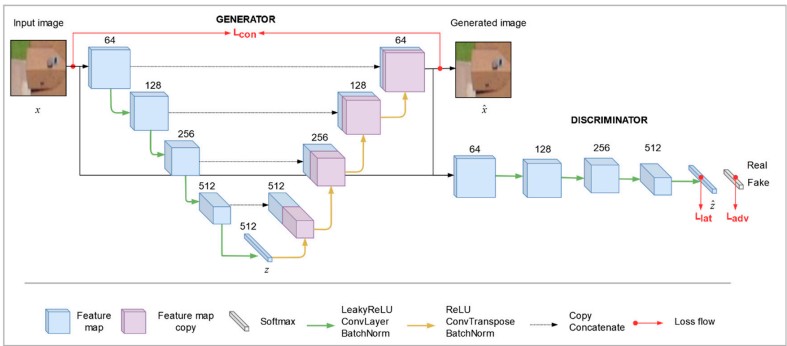

**Figure 1.** Skip-GANomaly architecture. Adapted from [24].

Skip-GANomaly makes use of three distinctive losses to guide its training, called the latent loss ($L_{lat}$), the adversarial loss ($L_{adv}$) and the contextual loss ($L_{con}$). $L_{adv}$ accounts for the correctness of the classification (fake or real). $L_{con}$ accounts for the generated image, and steers the model to create fake images that are contextually sound, i.e., images that look realistic. $L_{lat}$ is a loss that steers the encoders inside the Generator and Discriminator to create similar representations of the image latent vector $z$ [24]. Each loss contributes to the overall loss according to their corresponding weight (w). The losses are described in the following equations:

$$L_{adv} = \|f(x) - f(\hat{x})\|_2 \tag{1}$$

where,

$$f(.) = \mathbb{E}_{x \sim p_x} [\log D(.)] \tag{2}$$

$$L_{con} = \|x - \hat{x}\|_1 \tag{3}$$

$$L_{lat} = \|z - \hat{z}\|_2 \tag{4}$$

The overall loss is described as follows:

$$L = w_{adv}L_{adv} + w_{con}L_{con} + w_{lat}L_{lat} \tag{5}$$

Several hyper-parameters influence the performance of the model. Besides the general parameters such as batch size, learning rate or decay rate, model specific parameters include loss weights, the size of the latent vector z, and the number of encoder layers inside the Generator and Discriminator. Details on how these parameters are tuned can be found in Section 3.4.

A modification was made to the network. In the original network, after each epoch of training, the Area Under the Curve (AUC) score was calculated using the validation dataset. After training finished, a model for inference was selected based on the epoch in which it obtained the highest AUC score [24]. This makes the original implementation not a truly unsupervised approach, since a validation dataset is still required (i.e., examples of damage are still needed). Therefore, we choose to save the best performing model when the lowest Generator loss was found. This ensures that the model is chosen that is best able to generate fake images, which is the main principle of Skip-GANomaly. We verified that this approach yielded performance comparable to the original implementation, without the need of annotated test samples.

During inference, each image is classified as either damaged or undamaged by obtaining anomaly scores. Per-pixel anomaly scores are derived by simple image differencing between the input and the generated image. Each corresponding channel is subtracted from each other and averaged per pixel to obtain per-pixel anomaly scores. An image anomaly score is obtained by averaging the per-pixel anomaly scores. The closer to one, the higher the probability that the image is anomalous. After obtaining anomaly scores for all test samples, a classification threshold was determined in order to classify the images. This threshold is characterized as the intersection between the distribution of anomaly scores of normal and abnormal samples. Any sample with an anomaly score below the threshold was classified as normal and any value above the threshold as abnormal. Ideally, a model with a high descriptive value should result in non-overlapping distributions of the normal and abnormal samples with a clear threshold.

Finally, alterations and additions were applied to Skip-GANomaly in an attempt to boost results for the satellite imagery dataset. First, with the idea of weighing the generation of building pixels more than other pixels, we attempted to direct the attention of Skip-GANomaly by adding building masks as input in an approach similar to the one described in [48]. Furthermore, with the idea of utilizing the building information in the multiple epochs, similar to the approach described in [16], we stacked pre- and post-imagery into a 6-channel image and implemented an early, late or full feature fusion approach. These additions only provided marginal improvements. Our findings for stacking pre-

and post-imagery were in line with those found in [29]. The goal of this study was to investigate the applicability of ADGANs for building damage detection. Considering that improvements of the model were beyond our scope of work and only marginal, these lines of investigation were not explored any further and the original implementation was maintained.

### 3.2. Data

As mentioned earlier, a satellite and an UAV dataset were used in this research. This section will describe both datasets.

#### 3.2.1. xBD Dataset

We made use of the xBD satellite imagery dataset [49]. It was created with the aim of aiding the development of post-disaster damage and change detection models. It consists of 162.787 pre- and post-event RGB satellite images from a variety of disaster events around the globe. These include floods, (wild)fire, hurricane, earthquake, volcano and tsunami. The resolution of the images is 1024 × 1024, the GSD ranges from 1.25 m to 3.25 m and annotated building polygons were included. The original annotations contained both quantitative and qualitative labels: 0—no damage, 1—minor damage, 2—major damage and 3—destroyed [50]. The annotation and quality control process is described in [50]. The dataset contained neither structural building information nor disaster metrics such as flood levels or peak ground acceleration (PGA). Figure 2 shows example pre- and post-event images of a location where a volcanic eruption took place. The color of the building polygons indicates the building damage level. For our purpose, all labels were converted to binary labels. All images with label 0 received the new label 0—undamaged, and the ones with label 1, 2 or 3 received the label 1—damaged. We note that even though damage is labelled under the umbrella-label of the event that caused it, damage is most often induced by secondary events such as for example debris flow, pyroclastic flow or secondary fires. For the sake of clarity, we will refer to the umbrella-label when referring to induced damage. Example imagery of each event can be found in Figure 3a–j. This dataset was used in the Xview2 challenge where the objective was to localize and classify building damage [51]. The ranked top-3 submissions reported amongst others an overall F1-score of 0.738 using a multi-temporal fusion approach [29].

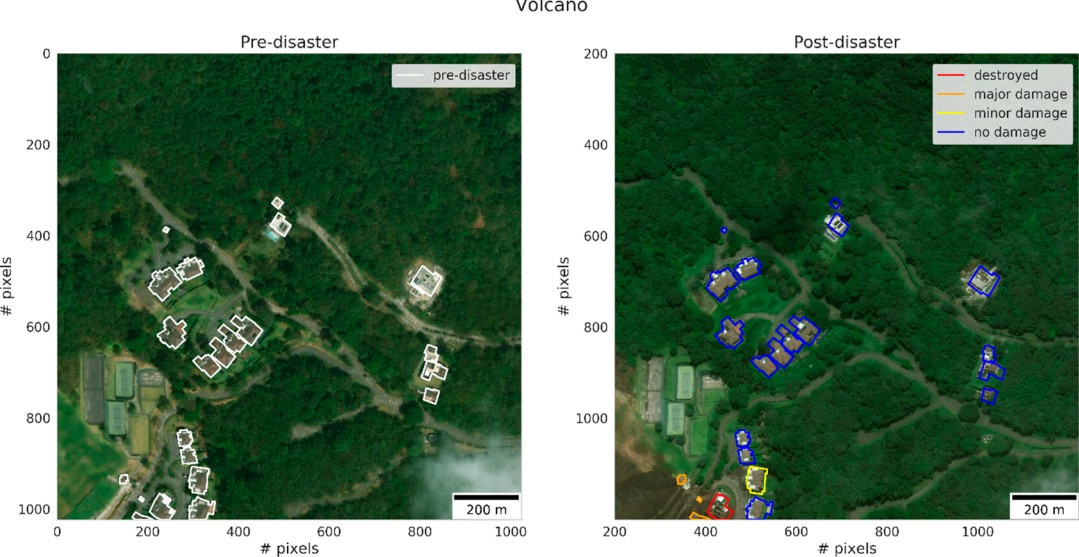

**Figure 2.** Example from the xBD dataset showing pre- and post-event satellite images from a location where a volcanic eruption took place. Several buildings and sport facilities are visible. The post-event image shows damage induced by volcanic activity. The buildings are outlined and the damage level is depicted by the polygon color. The scale bars are approximate.

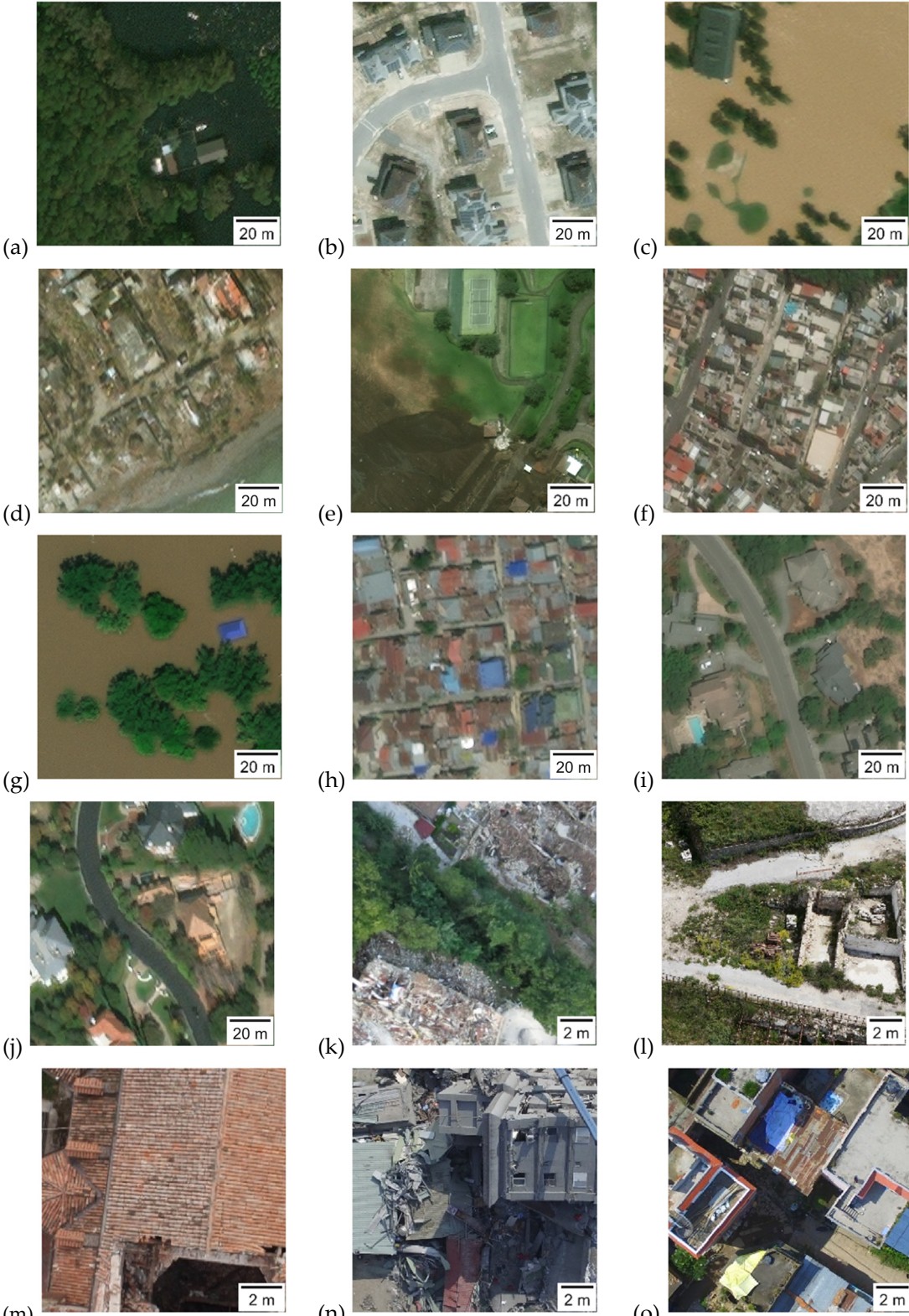

**Figure 3.** Examples of Satellite imagery used for testing: (**a**) Hurricane Florence (USA), (**b**) Hurricane Michael (USA), (**c**) Hurricane Harvey (USA), (**d**) Hurricane Mathew (Haiti), (**e**) Volcano (Guatemala), (**f**) Earthquake (Mexico), (**g**) Flood (Midwest), (**h**) Tsunami (Palu, Indonesia), (**i**) Wildfire (Santa-Rosa USA) and (**j**) Fire (Socal, USA). Examples of **UAV imagery** used for testing: (**k**) Earthquake (Pescara del Tronto, Italy), (**l**) Earthquake (L'Aquila, Italy), (**m**) Earthquake (Mirabello, Italy), (**n**) Earthquake (Taiwan) and (**o**), Earthquake (Nepal). The scale bars are approximate.

### 3.2.2. UAV Dataset

The UAV dataset was constructed manually from several datasets that depict the aftermath of several earthquake events. Examples can be found in Figure 3k–o. The UAV images were collected for different purposes and, therefore, the image resolution and the GSD vary and range around 6000 × 4000 pixels and from 0.02 to 0.06 m, respectively [13]. Moreover, the camera angle differed between nadir and oblique view. The UAV dataset contained no pre-event imagery and, therefore, the undamaged patches were obtained from undamaged sections in the images (see Section 3.3). Finally, the dataset contained neither structural building information nor PGA values.

### 3.3. Data Pre-Processing and Selection

Before the experiments were executed, the datasets were first treated to create different data-subsets. This section describes the different data treatments, while the next section describes how they were used in different experiments. The data treatments can be summarized into three categories: (i) varying patch size, (ii) removal of vegetation and shadows, and (iii) selection of data based on location or disaster type. For each dataset, we experimented with different cropping sizes. The rationale behind this step was that the larger the image, the more area is covered. Therefore, especially in satellite imagery where multiple objects are present, larger images often contain a high visual variety. As explained earlier, the Generator attempts to learn the image distribution, which is directly influenced by the visual variety contained in the images. When the learned image distribution is broad, a building damage has more chance to fall within this distribution, resulting in a reconstructed image that closely resembles the input image. The resulting distance between the input and generated images would be small and, therefore, the sample is expected to be misclassified as undamaged. We expected that restricting the patch size creates a more homogeneous and less visually varied scene. Especially cropping images around buildings would steer the Generator to learn mainly the image distribution of buildings. Therefore, any damage to buildings was expected to fall more easily outside the learned distribution, resulting in accurate damage detections and thus an increase in true positives.

The satellite imagery was cropped into patches of 256 × 256, 64 × 64 and 32 × 32 (Figure 4). By dividing the original image in a grid of four by four, patches of 256 × 256 could be easily obtained. However, the visual variety in these patches was likely still high. Smaller sizes of 64 × 64 or 32 × 32 would reduce this variety. However, simply dividing the original image systematically into patches of 64 × 64 or 32 × 32 resulted in a large amount of training patches that did not contain buildings. These patches did not contribute to learning the visual distribution of buildings. Therefore, the building footprints were used to construct 32 × 32 and 64 × 64 patches only around areas that contained buildings. To achieve this, the central point of each individual building polygon was selected and a bounding box of the correct size was constructed around this central point. We note that in real-world application, building footprints are not always available; however, this step is not necessarily required considering that it only intends to reduce the number of patches containing no buildings, even though there are various ways to derive building footprints. Open source repositories such as OpenStreetMap provide costless building footprints for an increasing number of regions, and supervised or unsupervised deep learning are proficient in extracting building footprints from satellite imagery [52–54]. Therefore, the proposed cropping strategy and subsequent training can be completely unsupervised and automated.

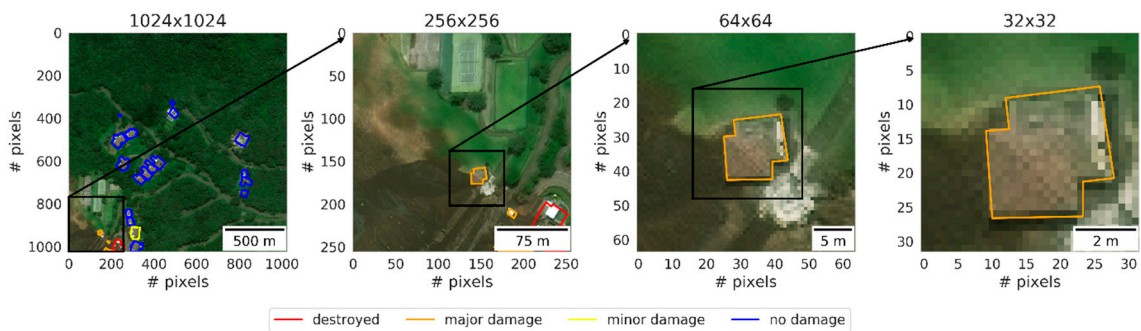

**Figure 4.** Illustration of different cropping strategies for the xBD dataset from the original patch size of 1024 × 1024 to 256 × 256, 64 × 64 and 32 × 32. The scale bars are approximate.

The UAV images were cropped in sizes of 512 × 512, 256 × 256 and 64 × 64. Larger patch sizes were chosen than for the xBD dataset to compensate for the difference in image resolution. More detail could be observed in larger sized UAV patches. Compare, for example, the amount of detail that can be observed in the smallest patches of Figures 4 and 5. Unlike for the xBD dataset, building footprints were not available. In general, building footprints for UAV imagery are difficult to obtain from open sources because, compared with satellite imagery, they are not necessarily georeferenced. Moreover, footprints would be difficult to visualize because of the varying perspectives and orientation of buildings in UAV imagery. Therefore, the 512 × 512 patches were extracted and labelled manually. Because of varying camera angles, patches displayed both facades and rooftops. Since no pre-event imagery was available, undamaged patches were obtained by extracting image patches from regions where no damage was visible. Binary labels were assigned to each image: 0—undamaged, or 1—damaged. The cropping strategy for the smaller sizes consisted of simply cropping around the center pixel (Figure 5).

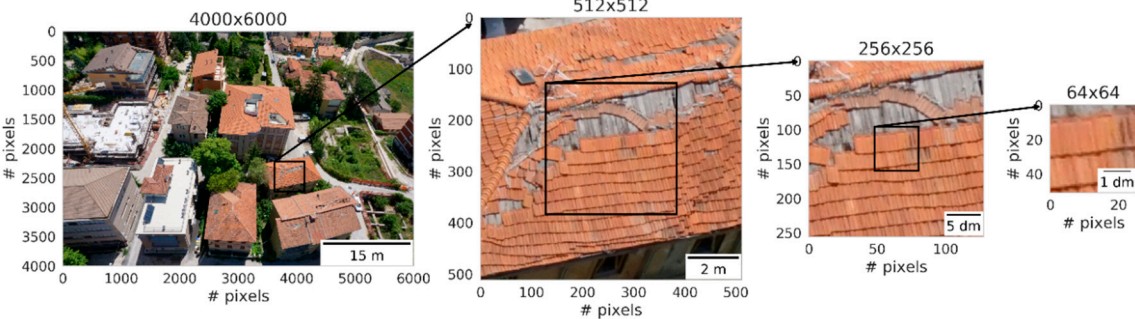

**Figure 5.** Illustration of cropping strategies for the UAV dataset from the original patch size of 4000 × 6000 to 512 × 512, 256 × 256 and 64 × 64. The scale bars are approximate and refer to the front of the scene.

Next, the cropped patches were pre-processed. In order to investigate how sensitive this method is against different pre-processing, images were removed from the dataset based on the presence of vegetation or shadows. Vegetation and shadows remain challenging in deep learning-based remote sensing applications. Shadows obscure objects of interest, but also introduce strong variation in illumination [55]. Depending on the varying lighting conditions, vegetation is prone to produce shadows and, therefore, varying Red, Green, Blue and illumination values [56]. Therefore, the image distribution learned by the Generator is expected to be broad. This means that any damage found on buildings is more likely to fall within this learned image distribution and, therefore, to be well reconstructed in the fake image. A well-reconstructed damage leads to lower anomaly scores, which is not the objective. We showed in [23] how removing these visually complex patches from the training set improve damage classification because the learned image distribution was expected to be narrower. Therefore, we created data subsets for training following the same procedure, using the

Shadow Index (SI; Equation (6)) and the Green–Red Vegetation Index (GRVI; Equation (7)) [57,58]. Images containing more than 75 or 10 percent vegetation and/or shadows, respectively, were removed from the original dataset. Using these datasets, we showed how increasingly stricter pre-processing, and thus decreasingly visually complex patches, influences performance. Removing images from a dataset is not ideal since it limits the practicality of the proposed methodology because it reduces the proportion of patches on which it can do inference (see Figure 6). The test dataset in the novegshad@10% data subset is 8 percent of the original test dataset. Therefore, we further experimented with masking the pixels that contain vegetation and shadow in an attention-based approach, as explained in Section 3.1. However, this was not considered as a further line of investigation since results did not improve.

$$SI = \sqrt{(256 - Blue) * (256 - Green)} \qquad (6)$$

$$GRVI = \frac{\rho_{green} - \rho_{red}}{\rho_{green} + \rho_{red}} \qquad (7)$$

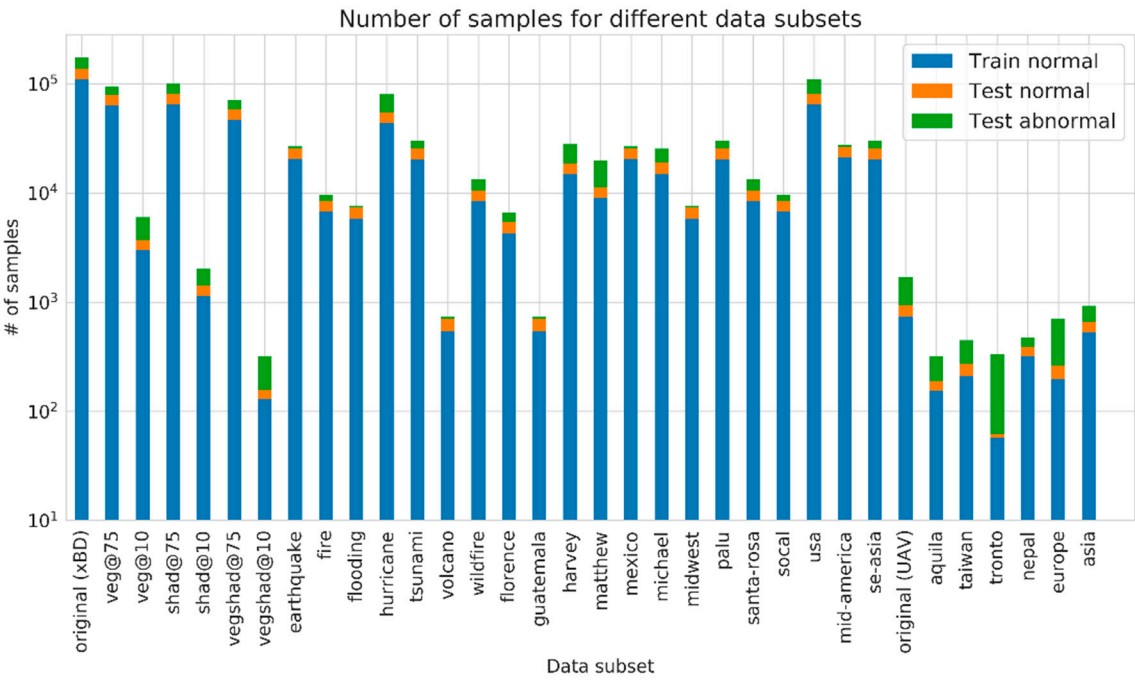

**Figure 6.** Number of samples in each data subset. Original refers to the complete un-preprocessed dataset. Y-axis is in log-scale.

Only the satellite patches of size $64 \times 64$ and $32 \times 32$ were pre-processed in this manner. Even though these sizes were already constrained to display maximum buildings and minimal surroundings using cropping techniques, some terrain and objects were often still present (see Figure 4). Satellite patches of size $256 \times 256$ were not pre-processed in this manner. Satellite images of this size usually contained more than 75 percent vegetation and/or shadow and, therefore, removing these images resulted in data subsets for training that were too small. UAV patches were also not pre-processed this way, since careful consideration was taken during manual patch extraction to ensure they do not contain vegetation or shadows.

Finally, selections of UAV and satellite patches were made based on the image location and the continent of the image location. Here the assumption was made that buildings were more similar in appearance if located in the same continent or country. Trained models were expected to transfer well to other locations if the buildings looked similar. Additionally, satellite patch selections were made based on the disaster type in order to investigate whether buildings affected by the same disaster type

could yield a high performance. Here we consider that end-users might already possess a database of pre-event imagery of the same disaster of different locations around the globe, while they are not in possession of pre-event imagery of the country or continent that appears similar to the location of interest. Table 1 shows a summary of all the resulting satellite and UAV data subsets.

**Table 1.** Overview of the data subsets used in this research. Data subsets were created based on (1) resolutions and (2) data selections, which include pre-processing (removal vegetation and/or shadows), disaster-event location, disaster-event continent and disaster-type. * Not for satellite patches of size 256 × 256.

| Dataset | Satellite (xBD) | | | | UAV | |
|---|---|---|---|---|---|---|
| **Resolutions** | 256 × 256 / 64 × 64 / 32 × 32 | | | | 512 × 512 / 256 × 256 / 64 × 64 | |
| **Category** | **Data Pre-Processing** | **Data Selection:** *Location* | **Data Selection:** *Continent* | **Data Selection:** *Disaster* | **Data Selection:** *Location* | **Data Selection:** *Continent* |
| | No vegetation (<75%) * | Guatemala (volcano) | North-America | Flood | Pescara del Tronto (Italy; earthquake) | Asia |
| | No vegetation (<10%) * | Florence (USA; hurricane) | Mid-America | Wildfire | Kathmandu (Nepal; earthquake) | Europe |
| | No shadow (<75%) * | Harvey (USA; hurricane) | South East Asia | Volcano | L'Aquila (Italy; earthquake) | South-America |
| | No shadow (<10%) * | Matthew (Haiti; hurricane) | | Hurricane | Portoviejo (Ecuador; earthquake) | |
| ← **Category values** | No vegetation and shadow (<75%) * | Michael (USA; hurricane) | | Earthquake | Mirabello (Italy; earthquake) | |
| | No vegetation and shadow (<10%) * | Mexico City (Mexico; earthquake) | | Tsunami | Taiwan (China; earthquake) | |
| | | Midwest (USA; flood) | | | | |
| | | Palu (Indonesia; tsunami) | | | | |
| | | Santa-Rosa (USA; wildfire) | | | | |
| | | Socal (USA; fire) | | | | |

Each data subset was divided into a train and test set. Figure 6 shows the sample size of each subset. The train-set only consisted of undamaged images, and the test set contained both undamaged and damaged samples. For the satellite imagery, the undamaged samples in the train set came from the pre-event imagery, whereas the undamaged samples in the test set came from the post-event imagery. For the UAV imagery, the undamaged samples both came from the post-event imagery. The samples were divided over the train and test set in an 80 and 20 percent split. The original baseline dataset denotes the complete UAV or complete satellite dataset.

We note that the UAV dataset size was relatively low. However, the authors of [44] found that the ability of an ADGAN to reproduce normal samples was still high when trained on a low amount of training samples. We verified that low number of samples had no influence on the ability of Skip-GANomaly to produce realistic output imagery, and thus we conclude that the Generator was able to learn the image distribution well, which was the main goal. For the reasons explained above, these numbers of UAV samples were deemed acceptable.

### 3.4. Experiments

The experiments were divided into two parts. Part one showed whether the method is applicable and/or sensitive to preprocessing. The experiments consisted of training and evaluating Skip-GANomaly models using the different pre-processed data subsets from Table 1, described in Section 3.3. Part two showed whether the method can be transferred to different geographic locations or disasters. The experiments consisted of training and testing a Skip-GANomaly model on the different location, continent and disaster data subsets. Each trained model, including the ones trained in part one, was cross-tested on the test set of each other data subset.

The training procedure maintained for part one and part two can be described as follows: A Skip-GANomaly model was trained from scratch using the train-set. Before training, the model was tuned for the hyper-parameters, $w_{adv}$, $w_{con}$, $w_{lat}$, learning rate and batch size, using grid-search. When the best set of hyper-parameters was found, the model was retrained from scratch using these parameter values for 30 epochs, after which it did not improve further. As explained earlier, using a modification, during training the best performing model was saved based on the lowest generator loss value. After training was completed, the model was evaluated on the test set. Training and evaluation ran on a desktop with a dual Intel Xeon Gold (3.6GHz) 8-cores CPU and a Titan XP GPU (12GB). Training for 30 epochs took approximately 8 hours using patches of $64 \times 64$ and a batch size of 64. Inference on a single image of $64 \times 64$ took approximately 3.9 ms.

For part two of the experiments, the transferability was analyzed by testing each of the previously trained models on the test set of all other data subsets. For example, all trained UAV models were evaluated on the UAV-imagery of all patch sizes from all locations and continents. All trained satellite models were evaluated on satellite-imagery of all patch sizes, from all locations and continents and from all pre-processing manners. To deal with different patch sizes, the images were up- or down-sampled during testing. Finally, transferability was not only analyzed intra-platform, but also cross-platform. This means that all models trained on different subsets of satellite imagery were also evaluated on the test set of all different subsets of UAV imagery and vice versa.

The F1-score, recall, precision and accuracy were used to describe performance. A high recall is important, because it shows that most instances of damage are indeed recognized as damage. In practice, this means that it can be trusted that no damage goes unnoticed. A high precision is also important, because it shows that from all the recognized instances of damage, most are indeed damage. Moreover, this means that it can be trusted that all the selected instances of damage are indeed damaged, and no time has to be spent on manually filtering out false positives. The F1-score represents the balance between recall and precision.

### 3.5. Comparison Against State-of-the-Art

In order to investigate how close our results can get to those of supervised methods, we compared the results of our experiments against results obtained using supervised deep learning approaches. In earlier work, we showed how unsupervised methods drastically underperformed compared to our method and, therefore, unsupervised methods such as One Class Support Vector Machine are left out of the comparison [23]. In order to make a fair comparison, we considered approaches that made use of a single epoch and, ideally, datasets that resemble ours in GSD, resolution, location and disaster-types. Therefore, we compared the results obtained using satellite-based models against the xView2 baseline model, and ranked competitors in the xView2 competition [29]. The xView2

baseline model first trained a U-Net architecture to extract building polygons. Afterwards, they used a Resnet50 architecture pre-trained on ImageNet to classify different degrees of building classifications. The ranked contenders [29] used a multi-temporal approach where both localization and classification was learned simultaneously by feeding the pre- and post-disaster images into two architectures with shared weights. The architectures consisted of ResNet50, which was topped with Feature Pyramid Heads, and were pre-trained on ImageNet. Finally, we compared the results obtained using UAV-based models with results obtained by [13]. Here, comparable UAV-images were used from post-earthquake scenes to train an adapted DenseNet121 network with and without fine-tuning. The authors carried out several cross-validation tests where each time a different dataset was used for testing, to investigate the influence of training data on performance.

## 4. Results

This section will describe the performance of Skip-GANomaly to detect building damage from satellite and UAV imagery. We show the more interesting results to avoid lengthy descriptions of all tests that were carried out. In addition, we present a closer look at the cases in which the model succeeded or failed to detect damage, and show how anomaly scores could be used to map damage. Additionally, the cross-test results are presented, which offer insight into the transferability of this method. Finally, a comparison between our results and supervised method is presented.

### 4.1. Performance of Skip-GANomaly on Satellite Imagery

First, we examined the performance of Skip-GANomaly on satellite imagery when using different data pre-processing techniques on the baseline dataset (all disasters combined). The main result showed that, especially when strict pre-processing was applied, e.g., removing all patches that contained more than 10 percent of vegetation or shadow (novegshad@10%), performance improved compared to baseline, although it only reached a recall value of 0.4 (Figure 7). A similar trend was found for aerial imagery in an earlier work [23]. Their performance improved the most when the novegshad@10% rule was applied. Finally, contrary to expectations and excluding the performance of novegshad@10% on 32 × 32 patches, no clear trend was observed for specific patch sizes. In some cases, the smaller sizes performed well and the larger size did not, and vice versa.

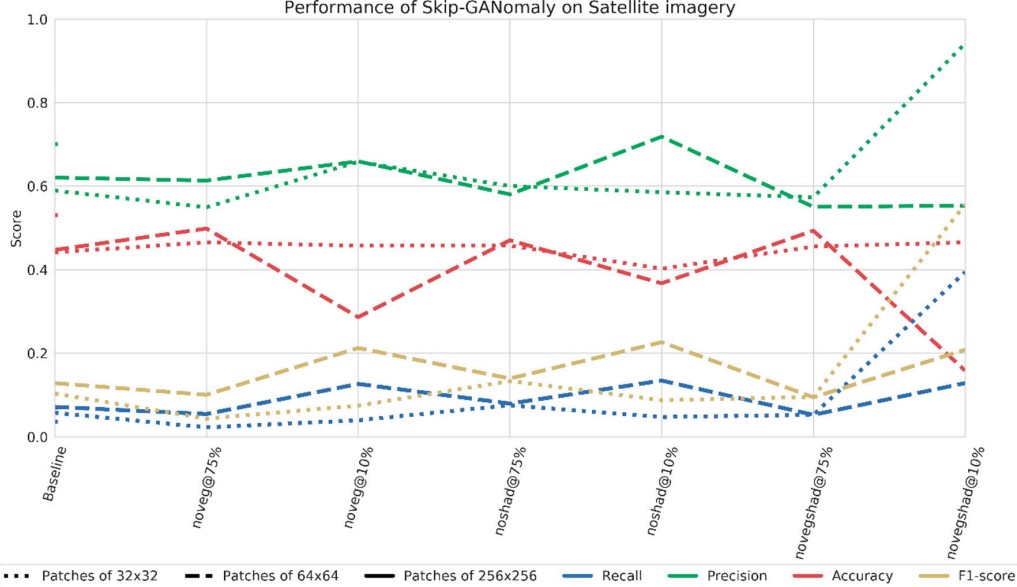

**Figure 7.** Performance of Skip-GANomaly on pre-processed satellite patches of size 256 × 256 (only baseline) 64 × 64 and 32 × 32.

Next, we examined the performance of Skip-GANomaly on satellite imagery when pre-selected by disaster type and without any pre-processing. Overall, we found that the performance of disaster-based models improved compared to baseline. Earlier, we found evidence that pre-processing improved performance. Therefore, in addition we tested the performance of disaster-based models when the training subsets of size $32 \times 32$ were pre-processed according to the novegshad@10% rule. The rule was not applied to subsets of size $256 \times 256$ or $64 \times 64$ because this resulted in subset sizes too small for training. The difference in performance is shown in Table 2. Again, we observed that performance improved for each individual disaster case.

**Table 2.** Difference in performance of Skip-GANomaly disaster-based models when trained on $32 \times 32$ satellite patches when (not) pre-processed based on the novegshad@10% rule. The grey background indicates the pre-processed values and bold values indicates which model performs best.

| Model | Pre-processed | Recall | Precision | F1-score |
|---|---|---|---|---|
| Earthquake | No | 0.110 | **0.212** | 0.022 |
| | Yes | **0.333** | 0.111 | **0.167** |
| Flooding | No | 0.455 | **0.555** | 0.500 |
| | Yes | **0.500** | 0.500 | **0.500** |
| Hurricane | No | 0.143 | 0.643 | 0.234 |
| | Yes | **0.325** | **0.662** | **0.436** |
| Tsunami | No | 0.040 | 0.365 | 0.073 |
| | Yes | **0.141** | **0.926** | **0.245** |
| Wildfire | No | 0.321 | 0.855 | 0.467 |
| | Yes | **0.778** | **0.989** | **0.871** |

We noted interesting differences between the performances of different disaster-based models (Table 2). Because a secondary damage induced by hurricanes is floods, it was expected that the performance for flood- and hurricane-based models would be comparable. However, this was not the case. In fact, it was observed that for the disaster types Hurricane and Tsunami (Table 2) and for the corresponding locations in Table 3, recall tended to be low compared to precision. We argue that this can be attributed to several reasons related to context, which will be explained in Section 4.3.

Finally, we examined the performance of Skip-GANomaly on satellite imagery when pre-selected based on location or continent location. In addition, the performance was examined when pre-processing according to the novegshad@10% rule was applied to patches of size $32 \times 32$. Again, we found that pre-processing improved the performance in a similar way, as was shown for the disaster-based models (Table 3).

## 4.2. Performance of Skip-GANomaly on UAV Imagery

Figure 8 shows the performance of UAV-based models. The main results show that the performance of UAV-based models was generally higher than that of the satellite-based models. Moreover, similar to the findings for satellite location-based models, we observed that the performance of UAV location-based models improved compared to baseline (all UAV-patches combined), with the exception of Asian locations (Nepal and Taiwan). Europe obtained a recall, precision and F1-score of 0.591, 0.97 and 0.735, respectively. As expected, UAV location-based models with similar building characteristics performed comparably. For example, models trained on location in Italy performed similarly (L'Aquila and Pescara del Tronto). This time, we did observe a pattern in performance of different patch sizes. Generally, models trained using the larger images size of $512 \times 512$ performed poorly, compared to models trained on smaller patch sizes. Especially for the Asian location-based models, the smaller sizes perform better. In the next section, we explain why context is likely the biggest influencer for the difference in performances.

**Table 3.** Difference in performance of Skip-GANomaly location-based models when trained on $32 \times 32$ satellite patches when (not) pre-processed based on the novegshad@10% rule. Locations that are not listed did not have sufficient training samples. The grey background indicates the pre-processed values and bold values indicates which model performs best.

| Model | Pre-processed | Recall | Precision | F1-score |
|---|---|---|---|---|
| Harvey (USA; hurricane) | No | 0.019 | 0.719 | 0.036 |
| | Yes | **0.198** | **0.800** | **0.317** |
| Matthew (Haiti; hurricane) | No | **0.144** | 0.625 | **0.234** |
| | Yes | 0.053 | **1.00** | 0.100 |
| Michael (USA; hurricane) | No | **0.291** | **0.800** | **0.427** |
| | Yes | 0.286 | 0.421 | 0.340 |
| Mexico City (Mexico; earthquake) | No | 0.055 | 0.002 | 0.005 |
| | Yes | **0.333** | **0.111** | **0.167** |
| Midwest (USA; flood) | No | 0.470 | 0.570 | 0.515 |
| | Yes | **0.750** | **0.600** | **0.667** |
| Palu (Indonesia; tsunami) | No | 0.099 | 0.393 | 0.158 |
| | Yes | **0.141** | **0.926** | **0.245** |
| Santa-Rosa (USA; wildfire) | No | 0.303 | 0.856 | 0.448 |
| | Yes | **0.684** | **0.985** | **0.807** |
| Socal (USA; fire) | No | 0.087 | 0.329 | 0.137 |
| | Yes | **0.538** | **0.667** | **0.596** |
| North-America | No | 0.099 | 0.718 | 0.175 |
| | Yes | **0.652** | **0.970** | **0.780** |
| Mid-America | No | 0.162 | 0.024 | 0.041 |
| | Yes | **0.333** | **0.100** | **0.154** |
| South East Asia | No | 0.031 | 0.366 | 0.058 |
| | Yes | **0.099** | **0.854** | **0.177** |

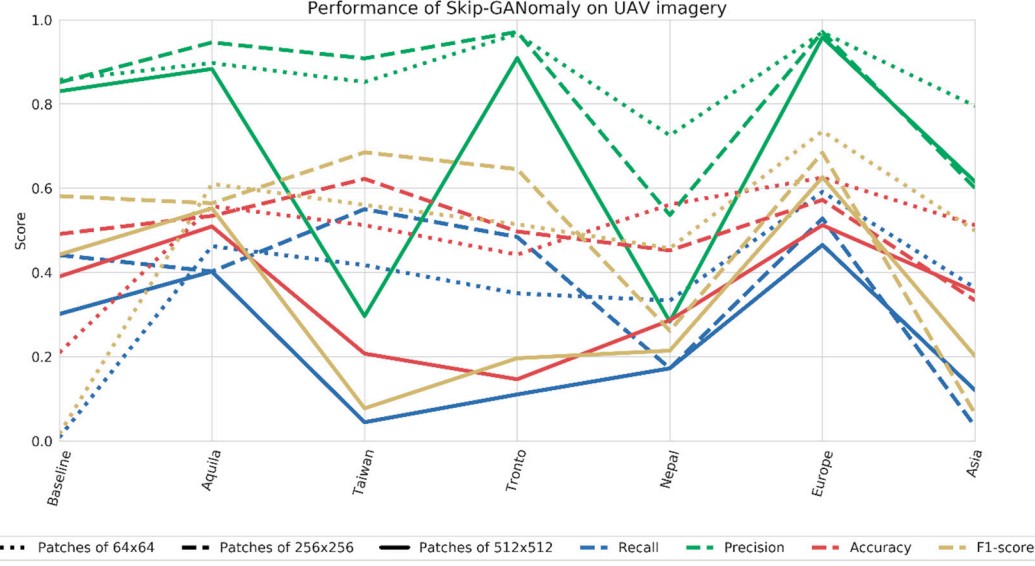

**Figure 8.** Performance of Skip-GANomaly on UAV imagery of size $512 \times 512$, $256 \times 256$ and $64 \times 64$ for different locations.

## 4.3. The Importance of Context

We investigated whether the characteristics of the different satellite data sources, especially for different disaster events, could explain why the method worked better for some disasters than the other. Certain disasters, such as floods or fires, induced both building damage and damage

to their surroundings. Other disasters such as earthquakes mainly induced damage to buildings only. Large-scale damage can be better detected from satellite imagery than small-scale damage, because satellite imagery contains inherently coarse resolutions. Most likely, the ADGAN is more efficient in detecting building damage from large-scale disasters by considering the surroundings of the building.

To investigate this idea, we aimed at understanding how the anomaly score distribution corresponds to the large-scale or small-scale damage pattern, by plotting the anomaly scores over the original satellite image. High anomaly scores on pixels indicate that the model considered these pixels to be the most anomalous. These pixels weigh more towards classification.

Figure 9 shows multiple houses destroyed by a wildfire. The image was correctly classified as damaged for all patch sizes. However, it seems that context contributed to the correct classification for the larger scale image (256 × 256), because the burned building surroundings resulted in high anomaly scores, whereas the buildings itself obtained lower scores. This suggested that, for this particular patch size, the surroundings have more discriminative power to derive correct classifications than the building itself. For smaller patch sizes, as explained in Section 3.3, the assumption was made that the smaller the patch size, the more adept the ADGAN would be in learning the image distribution of the building characteristics, instead of its surroundings. For the example in Figure 9, this seemed to hold true. In the patches of size 64 × 64 and 32 × 32, high anomaly scores were found all throughout the image, including the damaged building itself. This suggested that our assumption was correct. In short, the large-scale damage pattern of wildfire, plus the removal of vegetation, resulted in a high performing model. This example suggests that our method is capable of recognizing and mapping large-scale disaster induced damage.

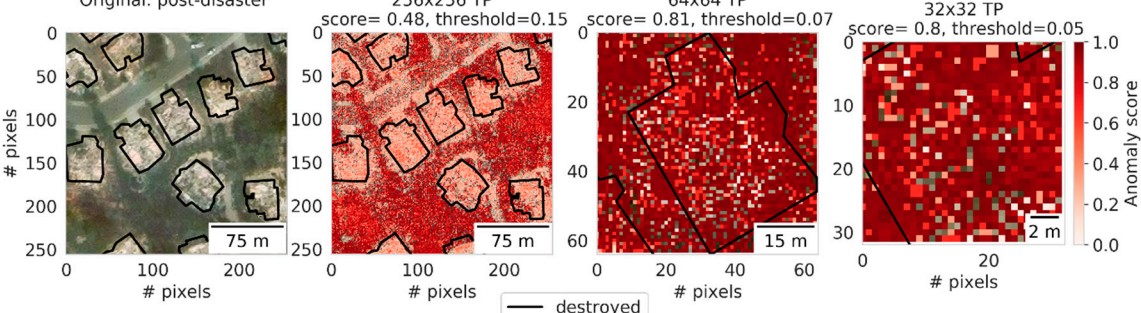

**Figure 9.** Post-wildfire satellite imagery from the USA showing multiple damaged buildings overlaid with anomaly scores and building polygon. The classification, classification threshold and anomaly scores are indicated (TP = True positive). The scale bars are approximate. High anomaly scores on burnt building surroundings and smaller patch sizes lead to correct classifications.

Another example of the influence of context can be seen in Figure 10. Here, a building was completely inundated during a flood event in the Midwest (USA), meaning that its roof and outline were not visible from the air. Even though, for all patch sizes, the image was correctly classified as damaged. We noticed how the anomaly scores were mostly located in the regions that were inundated. The anomaly scores on the unperceivable building itself naturally did not stand out. This suggests that the correct classifications were mostly derived from contextual information. No evidence of a better understanding of building damage could be observed for the smaller patch sizes (unlike for the wildfire example), because the building and its outline were not visible. Still, a correct classification was made due to the contextual information in these patch sizes. This example shows that the model was still adept in making correct inferences on buildings obscured by a disaster. Although removing vegetation generally improved classifications (Figure 7), for floods in particular performance only changed marginally (Table 2). Most of the flooded regions coincided with vegetated regions, which suggests

that the large-scale damage patterns induced by floods are of strong descriptive power and weigh up against the negative descriptive power of vegetation.

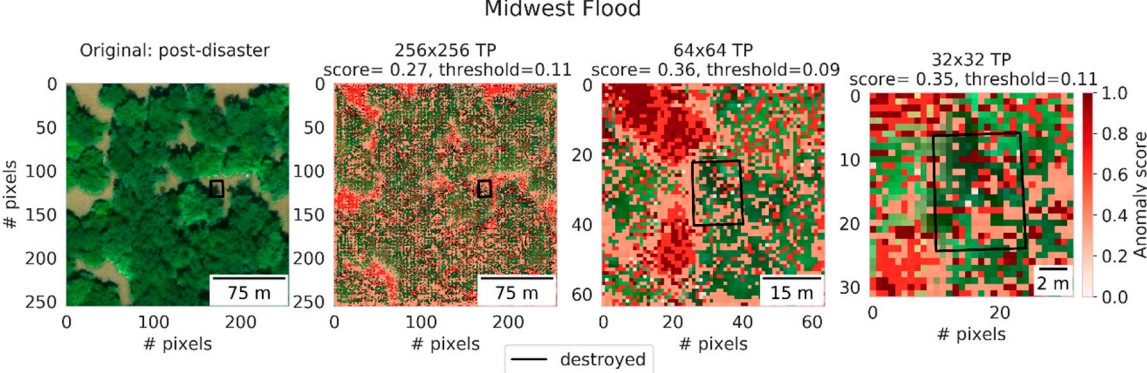

**Figure 10.** Post-flood satellite imagery from the USA showing a damaged building, overlaid with anomaly scores and building polygons. The classification, classification threshold and anomaly scores are indicated (TP = True positive). The scale bars are approximate. High anomaly scores on flooded areas resulted in correct classifications, regardless of patch size.

The examples above showed how context contributed to correct building damage classifications in cases where disasters clearly induced large-scale damage or changes to the surroundings. However, in cases where no clear damage was induced in the surroundings, context was shown to contribute negatively to classifications. Figure 11 shows an example of the Mexico earthquake event that spared multiple buildings. The patches of all sizes were wrongly classified as damaged, even though no large-scale damage was visible in its surroundings. The high anomaly scores in the 256 × 256 image were shown to exist all throughout the image, and to a lesser degree on the buildings itself. This example suggests that the context was visually too varied, which resulted in many surrounding pixels to obtain a high anomaly score. Moreover, unlike for the flood example, no clear damage pattern is present to counterbalance the visual variety of vegetation. As explained in Section 3.3, the variance could stem from vegetation. However, removing vegetation resulted in only modest improvements (Table 2). Therefore, the variation is also suspected to result from the densely built-up nature of the location. See for example Figure 3f, where a densely built-up region in Mexico without vegetation is visible. Even at the smaller patch sizes, the surroundings are varied, making it difficult to understand what is damaged and what is not. This example showed that context is less useful to derive earthquake-induced damage.

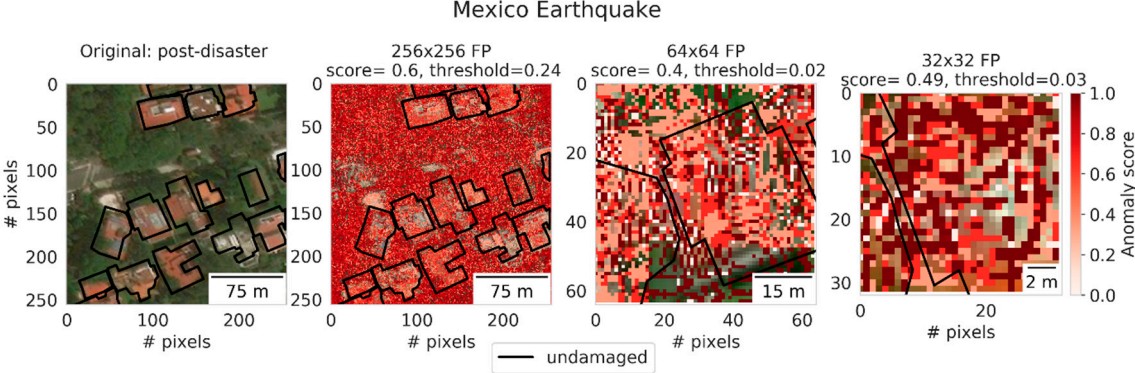

**Figure 11.** Post-earthquake satellite scene from Mexico showing undamaged buildings overlaid with anomaly scores and building polygons. The classification, classification threshold and anomaly scores are indicated (FP = False positive). The scale bars are approximate. High anomaly scores induced by varying building surroundings resulted in false positives, regardless of the patch size.

We argue that the examples shown so far could explain why recall for hurricane- and tsunami-based models was low (Tables 2 and 3). Where for flood events damage largely coincided with homogeneous flood patterns, the damage pattern for hurricanes and tsunamis was heterogeneous (Figure 3a–d,h). Large-scale floods and small-scale debris are the main indicators of damage. In addition, the locations suffer from a mixture between dense and less dense built-up areas. We suspect that when a mixture of damage patterns and inherently heterogeneous built-up area is present, lower performance and therefore a low recall value can be expected.

The examples shown above for satellite imagery can be used to illustrate the observed difference in performance for UAV location-based models, where smaller patch sizes were shown to perform better, particularly for the Asian locations (Nepal and Taiwan). Similar to the Mexico earthquake, Nepal and Taiwan are densely built-up. Moreover, the earthquakes in Nepal and Taiwan did not induce large-scale damage in the surroundings, meaning that the surroundings do not carry any descriptive power. However, unlike the Mexican imagery, due to the larger GSDs retained in the UAV imagery, reducing the patch size does result in less visual variety retained in smaller images. Therefore, similar to the wildfire event, and according to our previously stated assumption, smaller patch sizes result in a better understanding of the image distribution of buildings. Therefore, smaller images obtained higher anomaly scores on the building itself, instead of its surroundings, leading to performance improvements. See, for example, Figure 12.

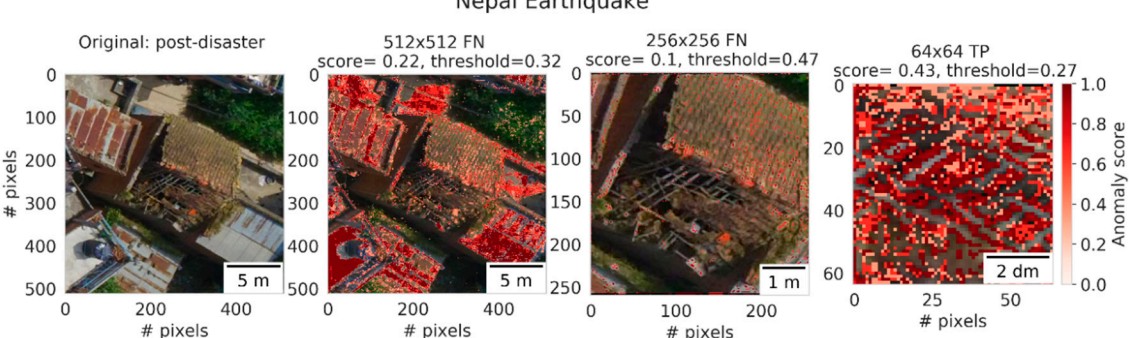

**Figure 12.** Earthquake UAV scene from Nepal showing a damaged building overlaid with anomaly scores. The classification, classification threshold and anomaly scores are indicated (TP = True positive, FN = False Negative). The scale bars are approximate. The $64 \times 64$ patch yielded a correct classification due to a better understanding of the building image distribution.

In summary, the presented examples showed how context is important for damage detections. Particularly, we found that both the scale of the induced damage and the GSD of the imagery decide whether context plays a significant role. One added benefit observed from these results is that in cases where context does play a role, the derived anomaly maps can be used to map damage. These damage maps are useful to first responders in order to quickly allocate relief resources.

*4.4. Cross-Tests*

Finally, we evaluated the performance of trained UAV- and satellite-based models on the test set of other data subsets to find out how transferable our proposed method is.

First, we highlight a couple of general findings for satellite-based cross-tests. In general, a heterogeneous pattern in performance was found when a trained satellite-based model was tested on the test-set of another satellite data subset. In some cases performance increased compared to the dataset on which it was trained, while for others it decreased. Overall, no noteworthy improvements in performance were observed. Second, we observed that satellite-based models did not transfer well to image datasets that had different patch sizes than the ones on which they were trained. For example, satellite-based models trained on patches of size $32 \times 32$ performed worse when tested on patches of

other sizes. This could be caused by context and GSD, as explained in Section 4.3. Finally, in some cross-tests, instead of the model, the dataset seemed to be the driving factor behind systematically high performances. For example, the flooding data subset yielded on average a F1-score of 0.62 for all patch sizes combined. This was systematically higher than for all other datasets. We expect that these results were driven by the flood-induced damage pattern, as explained in Section 4.3.

Next, we examined the cross-test results for UAV-based models. Again, general findings are highlighted. First, we observed that UAV-based models trained on specific patch sizes transferred well to patches of other sizes. A model trained on $32 \times 32$ patches from L'Aquila performed equally well on $64 \times 64$ patches from L'Aquila. We expect that this can be explained by the level of detail that is retained in UAV patches when the image is up- or down sampled during inference. Second, contrary to the findings for satellite imagery, no dataset seemed to drive the performance of datasets in a specific way.

Finally, we examined cross-platform transferability where we observed the performance of trained UAV or satellite models that were tested against the test set of either UAV or satellite imagery. The immediate observation is that the proposed method cannot be transferred cross-platform.

In summary, the models showed to transfer well if the test buildings look similar to the ones on which they were trained. Contrary to the desired outcome, transferability to buildings that are different in style was not unilaterally shown. We argue that the range of building appearances is yet too large for the Generator to learn an image distribution that is narrow enough to distinguish damaged buildings. Moreover, it is learned that no assumption can be made on the visual likeliness of buildings based on geographic appearance. However, we argue that transferability is less of an issue compared to traditional supervised methods considering that our method can be trained using pre-event data for the location of interest, which is often easily achieved.

### 4.5. Comparison Against State-of-the-Art

Table 4 shows how our proposed method compares against other state-of-of the art methods. Before comparing results for satellite-based models, we note that the xView2 baseline and the ranked contenders scores in the first and second row were based on the combined localization and multi-class classification F1-scores [29]. The third row shows the classification F1-score of the ranked contenders [29]. Our score is based on the binary classification F1-score. Because the supervised approaches considered all disasters at once, our reported F1-score is the average of F1-scores obtained using the pre-processed methods using patches of $32 \times 32$, which were reported in Table 2. Our method improved on the xView2 baseline but performed lower than the supervised method of the ranked contenders. Considering that our method is unsupervised, uses only pre-event imagery, makes no use of building footprints during damage detection training and is not pre-trained to boost performance, our results are encouraging and show that reasonable results can be obtained with minimal effort.

**Table 4.** Performance differences between our proposed method and comparable supervised CNN approaches.

|  | Method | Recall | Precision | F1-score |
|---|---|---|---|---|
| **Satellite** | Supervised—*localization and classification* (xView2 baseline [29]) | - | - | 0.265 |
|  | Supervised—*localization and classification* (Ranked contenders [29]) | - | - | 0.741 |
|  | Supervised—*classification* (Ranked contenders [29]) | - | - | 0.697 |
|  | Ours—*classification* | - | - | 0.444 |
| **UAV** | Supervised—*no fine-tuning* [13] | 0.538–0.814 | 0.741–0.934 | 0.623–0.826 |
|  | Supervised—*fine-tuning* [13] | 0.651–0.969 | 0.803–0.933 | 0.725–0.915 |
|  | Ours | 0.591 | 0.97 | 0.735 |

Before comparing results for UAV-based models, we note that the supervised results are reported as ranges, considering how different cross validation experiments were carried and reported in the original paper [13]. Our results are derived from our best performing European-based model using patches of 64 × 64 (Figure 8). Our results perform on par with the supervised approach without fine-tuning. In fact, our precision value exceeds the highest precision value found by the supervised methods. Our F1-score is constantly within the range of values found for the supervised method. Our recall score comes close to the lower range of recall values obtained using supervised methods. This suggested that, as can be expected, supervised methods outperform our unsupervised method. Moreover, it suggests that having examples of damage of interest are of benefit to recall values. Nonetheless, our method is more practical considering that it is unsupervised, and only requires single epoch imagery and, therefore, can be directly applied in post-disaster scenarios.

Finally, we note how the recall values for both the supervised and our method were always lower than the precision values, suggesting that building damage detection remains a difficult task despite using supervised methods.

## 5. Discussion

### 5.1. Applicability and Sensitivity of Skip-GANomaly

First, the applicability and sensitivity of satellite-based models are discussed. Satellite imagery is generally a difficult data type for unsupervised damage detection tasks due to their visual complexity, which is subject to temporal variety, and due to large GSDs, which make it difficult to detect detailed damage. This difficulty is reflected by the baseline results obtained using our proposed method. However, we showed how performance can be improved. In line with the results found for aerial imagery [23], reducing the complexity of the baseline dataset, e.g., removing vegetation and shadowed patches, improved performances, especially for the 32 × 32 novegshad@10% based model. Cropping the training patch sizes in order to reduce the visual complexity did not always yield better performances. We argued in Section 4.3 that context is suspected to play a role. Comparing the performance of our best scoring pre-processed model, a F1-score of 0.556, to the F1-score of 0.697 obtained by ranked contenders in the Xview2 contest for damage classification, our results are encouraging [29]. This is especially so considering that our method was unsupervised and only used a single epoch, whereas their methodology was supervised and multi-epoch.

We conclude that satellite-based ADGANs are sensitive to pre-processing, and reducing the complexity of the training data by applying pre-processing helps to improve performance. This finding is not necessarily novel. However, our specific findings on how reducing the complexity for specific disaster-types influences performance has provided insight on the importance of context, and allowed us to define practical guidelines that can be applied by end users. As an example, for disasters such as floods and fires, pre-processing is not strictly necessary to obtain good results. However, for other disasters, a downside of this method is that, once stricter pre-processing rules are applied, the numbers of samples on which the model can conduct inference declines. Future research can look into other ways to deal with vegetation or shadows. The idea of weighing these objects differently during training can be the focus, which, as explained in Section 3.1, was explored in early phases of this research.

Next, the applicability of UAV-based models will be discussed. We found that the UAV-based baseline model performed generally better than satellite-based baseline models. Location-based UAV models surpassed the performance of all satellite-based models, with the F1-score reaching 0.735. These results are satisfactory when compared to the F1-score of 0.931 obtained by [13], who used a supervised CNN to detect building damage from similar post-event UAV datasets. Again, considering our unsupervised and single-epoch approach, which only makes use of undamaged buildings for training, our method showed to be promising.

The importance of contextual information was explained in Section 4.3. We showed how flood or fire-induced building damage was likely deduced from contextual information, rather than from

the building itself. The contextual information has a negative influence towards the classification in disaster events where the damage is small-scale and the affected area is densely built-up. These findings suggest that, in practice, each event has to be approached case by case. However, we are able to provide broad practical guidelines: when the disaster has the characteristic of inducing large-scale damage to the terrain such as floods, the image training size can be $256 \times 256$.

Finally, we showed in Section 4.3 how detailed damage maps can be derived using simple image differencing between the original and generated image. As of yet, we are not aware of any other method that can produce both image classifications and damage segmentations without explicitly working towards both these tasks using dedicated supervised deep learning architectures and training schemes. Our method, therefore, shows a practical advantage compared to other methods.

Future work can focus consider the following: the rationale behind our sensitivity analysis was that reducing the visual information being fed to the Generator steers the Generators inability to recreate damaged scenes, which in turn helps the Discriminator distinguish fake from real. As an extra measure, future work can focus on strengthening the discriminative power of the Discriminator earlier on in the training process, by allowing it to train more often than the Generator, thus increasing its strength while not limiting the reconstructive capabilities of the Generator. Future work can also investigate the potential of alternative image distancing methods to obtain noiseless anomaly scores. The log-ratio operator for example, often used to difference synthetic aperture radar (SAR) imagery, considers the neighborhood pixels to determine whether a pixel has changed [59]. It is expected that such a differencing method lead to a decrease of noisy anomaly scores, and thus a better ability to distinguish between anomalous and normal samples.

*5.2. Transferability*

In general, a heterogeneous performance was observed for satellite-based models when tested on test-sets of other satellite sub-sets. Performance fluctuated for the different datasets and regardless of whether the model tested well on the dataset for which it was trained. This suggests that satellite-based models do not transfer well to other geographic locations or other typologies of disasters. This finding is in contrast with one specific finding from our preliminary work. There, we found that aerial-based models, trained on patches that were pre-processed according to the novegshad@10% rule, transferred well to other datasets [23]. We did not observe the same for the satellite-based model novegshad@10%. A possible explanation can be that the novegshad@10% model was not able to find the normalcy within other datasets, because the amount of training samples is small (see Figure 6). Therefore, the learned image distribution is too narrow. This could have led to an overestimation of false positives once this model was tested on other datasets.

Contrastingly, a homogeneous performance was observed for UAV-based models when tested on test-sets of other UAV sub-sets. Consistent performance was observed when models were tested on different datasets or different patch sizes. In addition, the model performance stayed high if the performance was high for the dataset on which it was trained. Specifically, we found that models transferred well if the buildings on which the model was tested looked similar to the buildings on which it was trained. For example, locations in Italy (L'Aquila and Pescara del Tronto) looked similar and were shown to transfer well. Locations in Asia (Taiwan and Nepal) looked very dissimilar in appearance and did not transfer well. Similar conclusions for the transferability of Italian locations were found in [13]. In line with the conclusion drawn in [13], we agree that the transferability of a model depends on whether the test data resemble the data on which it was trained. A model that cannot find the normalcy in other datasets is likely to overestimate damage in this dataset. Therefore, our previously stated assumption that buildings in the same geographic region look alike is not always valid. In future approaches, attention should be given to how geographic regions are defined. Categorizing buildings not based on the continent in which they are located, but on lower geographic units such as municipalities or provinces, might lead to a better approximation by the AGDAN of what constitutes a normal building in that geographic unit.

### 5.3. Practicality in Real-world Operations

The general conclusion is drawn that ADGANs can be used for damage detection from satellite images on the condition that the imagery is pre-processed to contain minimal vegetation and shadows. Considering how pre-processing is largely automated, this step is not a limitation. Nonetheless, cases were found where models yielded high performance, regardless of the presence of vegetation and shadows. The performance of satellite-based models trained on original imagery from flood and fire disasters was high and, therefore, these datasets do not have to be pre-processed, thus saving time.

We showed that damage maps could be constructed in the cases where context provides a significant contribution. These show, in detail, where damage is located. During inference, these maps can be created instantly, and they can therefore provide valuable information in the post-disaster response and recovery phase.

As stated in the introduction, a main limitation of UAV-based models is that UAV-based imagery needs to be collected in the pre-event stage. Considering how UAV-imagery collection is still a human-driven task, this might be difficult to achieve. However, the advantage is that data acquisitions can take place any time during the pre-event stage. Therefore, practical advice to end-users who wish to apply this methodology is to collect UAV-imagery of buildings in the pre-event stage in advance.

A final note of consideration is the following: the assumption is made that the normal dataset is free of anomalies. However, day-to-day activities such as constructions can result in visual deviations from normal that are not strictly damage [45]. In practice, care has to be taken to make the distinction between what is damaged and what is simply an anomaly.

## 6. Conclusions

In this paper, we proposed the use of a state of the art ADGAN, Skip-GANomaly, for unsupervised post-disaster building damage detection, using only imagery of undamaged buildings from the pre-epoch phase. The main advantage of this method is that it can be developed in the pre-event stage and deployed in the post-event stage, thus aiding disaster response and recovery. Special attention was given to the transferability of this method to other geographic regions or other typologies of damage. Additionally, several Earth observation platforms were considered, since they offer different advantages for data variety and data availability. Specifically, we investigated (1) the applicability of ADGANs to detect post-disaster building damage from different remote sensing platforms, (2) the sensitivity of this method against different types of pre-processing or data selections, and (3) the generalizability of this method over different typologies of damage or locations.

In line with earlier findings, we found that the satellite-based models were sensitive against the removal of objects that contained a high visual variety: vegetation and shadows. Removing these objects resulted in an increase in performance compared to the baseline. Additionally, in order to reduce the visual variety in the original images, experiments were conducted with varying image patch sizes. No clear difference in performance of different patch sizes was observed. UAV-based models yielded high performance when detecting earthquake-induced damage. Contrary to satellite-based models, UAV-based models trained on smaller patches obtained higher scores.

UAV imagery contained small GSDs and showed damage in high detail. Therefore, models based on UAV-imagery transferred well to other locations, which is in line with earlier findings. Models based on satellite-imagery did not transfer well to other locations. The results made it evident that image characteristics (patch size and GSD), and the characteristics of the disaster induced damage (large-scale and small-scale), play a role in the ability of satellite-based models to transfer to other locations.

Compared to supervised approaches, the obtained results are good achievements, especially considering the truly unsupervised and single-epoch nature of the proposed method. Moreover, the limited time needed for training in the pre-event stage and for inference in the post-event stage (see Section 3.4) make this method automatic and fast, which is essential for its practical application in post-disaster scenarios.

**Author Contributions:** Conceptualization, S.T. and F.N.; Analysis, S.T.; Methodology, S.T. and F.N.; Supervision, F.N., N.K. and G.V.; Original draft, S.T. and F.N.; Review and editing, S.T., F.N., N.K. and G.V. All authors have read and agreed to the published version of the manuscript.

**Funding:** Financial support has been provided by the Innovation and Networks Executive Agency (INEA) under the powers delegated by the European Commission through the Horizon 2020 program "PANOPTIS–Development of a decision support system for increasing the resilience of transportation infrastructure based on combined use of terrestrial and airborne sensors and advanced modelling tools", Grant Agreement number 769129.

**Conflicts of Interest:** The authors declare no conflict of interest.

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
