# Peer review of "Post-Disaster Building Damage Detection from Earth Observation Imagery Using Unsupervised and Transferable Anomaly Detecting Generative Adversarial Networks"

_remotesensing, doi:10.3390/rs12244193_

Round 1

Reviewer 1 Report

The title of the manuscript (MS) deals with "Post-Disaster Building Damage Detection from Earth Observation Imagery using Unsupervised and Transferable Anomaly Detecting Generative
Adversarial Networks". The topic of this manuscript is of interest and well written and I liked reading it, great job!

Just one comment.
Your conclusion is too long.
This should not a summary of your paper. Try to be more clear.
"Usually, the conclusion should be neither too short nor too long. When the conclusion is too long, it leaves the impression that you are disorganized, trying to fit in the material that should have been in the body of the essay".

Author Response

Dear reviewer,

Thank you for your compliment and your comment. We have revised the conclusion and moved several paragraphs on the practical usage of our work to a separate heading in the discussion. Please refer to section 5.3 and chapter 6.

Reviewer 2 Report

The work is very valuable from a scientific point of view and is well written. Some minor revisions are necessary before publication.

Introduction: A more in-depth discussion is needed with reference wo the typologies of damages that can be detected from the system (minor/major damages). This work is very valuable but it needs a direct connection with the loss assessment scenario investigations or with the empirical fragility/vulnerability estimation for system of building.

Some references can be done to a work that used directly the satellite data to provide this type of analysis:

Rota M, Penna A, Strobbia CL (2008) Processing Italian damage data to derive typological fragility curves. Soil Dyn Earth Eng 28(10–11):933–947

Rossetto T, Elnashai A (2003) Derivation of vulnerability functions for European-type RC structures based on observational data. Eng Struct 25(10):1241–1263

Charvet I, Ioannou I, Rossetto T, Suppasri A, Imamura F (2014) Empirical fragility assessment of buildings affected by the 2011 Great East Japan tsunami using improved statistical models. Nat Hazards 73(2):951–973

Miano, A., Jalayer, F., Forte, G., and Santo, A. (2020). Empirical fragility assessment using conditional GMPE-based ground shaking fields: application to damage data for 2016 Amatrice Earthquake. Bulletin of Earthquake Engineering, 1-31.

Introduction: Is there any relation in the capacity of catching the damage with the structural type of the building/infrastructure (masonry, reinforced concrete, bridge etc.)

Figure 2. Please distinguish Figure 2a from Figure 2b. This comment can be extended also to other Figures. What is happened to the other buildings not colored in Figure 2b?

Figure 3. I would increase a little bit the size of the Figures.

Section 3.3 Better clarify how the vegetation can alter the results.

Section 3.3 Same for the presence of shadows zones.

Table 1. Maybe a more in depth discussion on the dependence of the results from the type of actions is needed.

Figure 6. Please increase a little bit the size of the text in the bottom part of the Figure.

Reviewer 3 Report

The authors proposed a GAN-based method for detecting post-disaster building damages. Several concerns have to be clarified to make the manuscript get published:

  1. The proposed method needs a building footprints information. While the authors mention that some open source repositories do provide this information. However, it is not clear if they can be perfectly integrated with the proposed approach. More experiments should be done to verify this.
  2. The authors eliminate images containing more than 75 or 10 percent vegetation and/or shadows. However, we do not know the proportion of these images in the dataset. These images may still be worth analyzing.
  3. It is to be expected that with such strict preprocessing rules, performance will be improved as the authors eliminate those "hard data" that may often appear in real-world applications.
  4. The recall rate in table 3 is quite low. The authors should provide a fair discussion about the reason why.
  5. The authors use many methods to clean the dataset, which make the remaining data easier to process. However, these preprocessing methods may not be feasible in practice. In my opinion, a fair comparison should be made between other SOTA to prove the robustness of the proposed method.
  6. The authors do not mention how to get the pixel-wise anomaly scores. (Fig. 9-12)
  7. The authors do not made great modifications on ADGAN model so that the contribution is not significant.
  8. The authors claimed that the proposed method is unsupervised and transferable. However, the GAN model is unsupervised but the pre-processing are manual. Besides, the model is transferable if the locations have similar building styles, which is not surprising.

Reviewer 4 Report

This paper used the Anomaly Detecting Generative Adversarial Network (ADGAN) to accomplish the post-disaster building damage detection task. Although the whole idea is significant in engineering, the academic contribution is limited. I do not recommend the current version publish in Remote Sensing. Some detailed suggestions are displayed as follows.

  1. The main contributions and motivation are not clear. The authors only select an existing model to complete their task and report some results.
  2. Why do you choose ADGAN? There are many similar unsupervised models that can be used.
  3. The experiments are diverse and sufficient, however, I cannot find any comparisons. Some latest models should be chosen to compare with your idea.
  4. Please improve your work from the aspect of academics.

Round 2

Reviewer 3 Report

The authors have answered my questions and revised the manuscript. I have no further questions.

Author Response

Dear reviewer,

We are happy to hear that you think our revised manuscript has improved to your satisfaction. Thanks again for your questions and comments.

Best regards,

The authors

Reviewer 4 Report

All of the issues have been modified, and the quality of the current version is improved a lot. Only one minor suggestion, i.e., some latest literature related to building extraction in remote sensing could be cited for enriching your reference. For example,

Domain Adaptive Transfer Attack-Based Segmentation Networks for Building Extraction From Aerial Images;

Building Extraction of Aerial Images by a Global and Multi-Scale Encoder-Decoder Network;

... 

Author Response

Dear reviewer,

We are happy to hear that you think our revised manuscript has improved. Thank you for your final suggestions. We feel that the first paper is not entirely fitting our paper because although the paper makes use of generative adversarial networks to generate new samples to boost the performance of their building segmentation model, it deals with 1) generalizability instead of transferability and, 2) buildings segmentation rather than building damage segmentation.

However, the second recommended paper supports our statement that “… supervised and unsupervised deep learning are proficient in extracting building footprints from satellite imagery [52, 53]. We have added this reference to this sentence in section 3.3.:

“Open source repositories such as OpenStreetMap provide costless building footprints for an increasing number of regions, and supervised or unsupervised deep learning are proficient in extracting building footprints from satellite imagery [52-54].”

Thanks again for the questions, comments and suggestions.

Best regards,

The authors.